# Development and Challenges of Antimicrobial Peptides for Therapeutic Applications

**DOI:** 10.3390/antibiotics9010024

**Published:** 2020-01-13

**Authors:** Charles H. Chen, Timothy K. Lu

**Affiliations:** 1Synthetic Biology Center, Massachusetts Institute of Technology (MIT), Cambridge, MA 02139, USA; 2Synthetic Biology Group, Research Laboratory of Electronics, Massachusetts Institute of Technology (MIT), Cambridge, MA 02139, USA; 3Broad Institute of MIT and Harvard, Cambridge, MA 02142, USA; 4Department of Electrical Engineering and Computer Science, Massachusetts Institute of Technology (MIT), Cambridge, MA 02142, USA; 5Harvard-MIT Division of Health Sciences and Technology, Cambridge, MA 02139, USA; 6Department of Biological Engineering, Massachusetts Institute of Technology (MIT), Cambridge, MA 02142, USA

**Keywords:** antimicrobial peptides, antibiotics, antibiotic resistance, rational protein design, peptide therapeutics, FDA-approved peptides

## Abstract

More than 3000 antimicrobial peptides (AMPs) have been discovered, seven of which have been approved by the U.S. Food and Drug Administration (FDA). Now commercialized, these seven peptides have mostly been utilized for topical medications, though some have been injected into the body to treat severe bacterial infections. To understand the translational potential for AMPs, we analyzed FDA-approved drugs in the FDA drug database. We examined their physicochemical properties, secondary structures, and mechanisms of action, and compared them with the peptides in the AMP database. All FDA-approved AMPs were discovered in Gram-positive soil bacteria, and 98% of known AMPs also come from natural sources (skin secretions of frogs and toxins from different species). However, AMPs can have undesirable properties as drugs, including instability and toxicity. Thus, the design and construction of effective AMPs require an understanding of the mechanisms of known peptides and their effects on the human body. This review provides an overview to guide the development of AMPs that can potentially be used as antimicrobial drugs.

## 1. Introduction

In the past several decades, multidrug-resistant bacteria have rapidly spread, causing increases in nosocomial infections and in-hospital mortality, and posing a threat to global health [1,2,3,4]. Moreover, the discovery of new classes of antibiotics has slowed down since 1987 [5,6]. The lack of new discoveries might be prompted by the conservative way we have searched for antibiotics, or this field may be saturated [5,6]; in other words, we may have already discovered many of the large natural structures that have antimicrobial activity. With the rise of antibiotic resistance, our last lines of effective antibiotics are failing [7,8,9]. Antimicrobial peptides (AMPs), a ubiquitous part of the innate immune defense in all classes of life, have been widely studied and show potential as small molecule antibiotics [10,11,12].

## 2. FDA Drug Approvals and Databases

More than 3000 AMPs have been reported and characterized [13], but in their natural state, most are not suitable as drugs for human medicine. In fact, many of them failed prior to or during clinical trials [14]. To understand this problem, we looked into peptide drugs that have been approved by the U.S. Food and Drug Administration (FDA). Using the Therapeutic Proteins Database [THPdb, a subset of the FDA database (Drugs@FDA)], we analyzed all the peptide therapeutics so far approved by the FDA [15]. This database has a total of 852 peptide and protein therapeutics. Of these, 239 have been validated while the others are their derivatives and/or similar ingredients for therapeutics, and 27 of the 239 are small peptides (fewer than 50 amino acids). Six of the small peptides are AMPs: gramicidin D, daptomycin, vancomycin, oritavancin, dalbavancin, and telavancin (Figure 1).

Gramicidin D, first isolated from the soil bacterium *Bacillus brevis* and characterized in 1941, is a heterogeneous mixture of three pore-forming peptides: gramicidins A (80%), B (5%), and C (15%) [16,17,18,19,20,21]. Gramicidin D was approved by the FDA in 1955 as a constituent in Neosporin^®^ [22], a triple antibiotic ointment for treating bacterial conjunctivitis. Daptomycin is a 13-residue cyclic lipopeptide antibiotic that binds onto the bacterial cell membrane, aggregates, and disrupts the membrane [23,24]. Daptomycin (also known as LY146032 [25]) and its derivative Cubicin (manufactured by Cubist Pharmaceuticals, now Merck & Co.) were approved in 2003 by the FDA to treat or prevent infectious diseases [26]. Cubicin and its new formulation Cubicin RF, which can be directly injected into the body, are antibiotics used for the treatment of complicated skin and skin structure infections (cSSSI) and *Staphylococcus aureus* bloodstream infections. Oritavancin, dalbavancin (formerly BI-397), and telavancin are small lipoglycopeptide antibiotics derived from vancomycin (approved by the FDA as an oral solution in 1983). These lipoglycopeptides are more potent and bactericidal than their prototype vancomycin, and they are effective against vancomycin-resistant bacteria. They inhibit bacterial cell wall formation [27,28], and telavancin and oritavancin also disrupt bacterial cell membranes and affect membrane permeability [29,30]. Similar to Cubicin, the therapeutic products Orbactiv (oritavancin), Dalvance (dalbavancin), and Vibativ (telavancin) are being used for injection against cSSSI caused by *S. aureus*, as well as other Gram-positive bacterial infections, and were approved by the FDA in 2014, 2014, and 2009, respectively.

One FDA-approved AMP is not mentioned in the THPdb. Colistin, also known as polymyxin E, is an antibacterial cyclic lipopeptide that was approved by the FDA in 1962 and is manufactured by Endo Pharmaceuticals, Inc, Malvern, Pennsylvania, U.S.A. as the product Coly-Mycins (colistin sulfate). Colistin is composed of 10 amino acids (sequence: KTKKKLLKKT; MW = 1145) and one fatty acid (6-methyl octanoic acid). Six of the constituent amino acids are positively charged (e.g., lysine), while the hydrophobic content is 20%. Colistin is active against several Gram-negative bacteria, e.g., *Pseudomonas aeruginosa*, *Klebsiella pneumoniae*, and *Acinetobacter* spp.

Peptide stability is a key requirement for the use of peptides as drugs [31,32,33,34]. Nevertheless, the hormone insulin and its analogs, which are among the most well-known peptides, have a short elimination half-life (4-6 min) in the bloodstream. Insulin was the first genetically engineered peptide hormone and was approved by the FDA in 1982 for the treatment of diabetes [35,36]. The elimination half-life of FDA-approved AMPs is much longer than that of insulin [13,31,32,33,34]. Daptomycin, oritavancin, dalbavancin, telavancin, and colistin have elimination half-lives of 8–9 h, 14 days, 8 h, 195.4 h, and 5 h, respectively (that of gramicidin has not been determined). More broadly, the average elimination half-life of FDA-approved new drugs is 50 h (median = 9 h), and of FDA-approved small peptides (less than 50 amino acids) for therapeutic use is 37 h (median = 3 h) (Figure 2). Thus, most of the FDA-approved peptides included in this analysis are stable *in vivo*, probably because those that are not biologically stable are unsuitable as drugs [31,32,33,34]. More rigid peptide structures may extend the elimination half-life [37,38,39]. For example, cyclic lipopeptides (e.g., daptomycin and colistin) and cyclic lipoglycopeptides (e.g., vancomycin, oritavancin, dalbavancin, and telavancin) are more stable than their linear counterparts [40]. In addition, introducing non-canonical amino acids into the peptide sequence can prevent biological degradation by proteases and extend the elimination half-life [41].

## 3. Antimicrobial Peptide Database

To date, only seven small AMPs have been approved by the FDA, so we extended our study to other AMPs that are under development and listed in the Antimicrobial Peptide Database (APD). A total of 3156 AMPs is listed in the APD, most of which were discovered in nature [13]. An analysis of 2700 of the 3156 AMPs in the APD showed that these peptides all have different structures and sequence motifs, and because they have a broad spectrum, they can kill a range of pathogens [42,43]. Interestingly, one-third of the AMPs are derived from frogs [44]. The average length of peptides in the APD is 33 amino acids, the median length is 28 amino acids, and more than 90% of the peptides, known as small peptides, have no more than 50 amino acids (Figure 3A). The average hydrophobic content of the peptides is 54% (Figure 3B), and the mean peptide net charge is +3 (Figure 3C). About 45% of the peptides do not contain cysteine; 21% and 17% of them have two cysteines and six cysteines, respectively (Figure 3D), which reveals the potential of a disulfide bond formation between two cysteines.

In the APD, 1869 of the 2700 peptides (~70% of the database) are small cationic amphipathic peptides. However, of the FDA-approved AMPs, only one, colistin, is in this category of small cationic amphipathic peptides. Gramicidin is small (10 amino acids) and has a net charge of +2, but it contains eight hydrophobic residues and two positively charged lysine residues, which makes it a small cationic hydrophobic peptide. Many studies have proposed that membrane-active AMPs selectively target and disrupt anionic bacterial cell membranes using electrostatic interactions [45,46,47,48,49]. However, daptomycin, a small amphipathic peptide with a neutral net charge, deviates from this pattern. Because vancomycin, oritavancin, dalbavancin, and telavancin are lipoglycopeptides, they are not included in the APD, which comprises only peptides and lipopeptides.

## 4. Current Development of Peptide Drugs

We further analyzed FDA-approved small peptide (less than 50 amino acids) therapeutics from the past 20 years (total 57 drugs; Table 1) using Drugs@FDA (http://www.fda.gov/drugsatfda) and DrugBank [50]. Details regarding newly approved compounds (submission classification: Type 1—New Molecular Entity) were extracted from Drugs@FDA, and the data were further confirmed by DrugBank. A total of 555 new molecules were approved and commercialized between January 1999 and December 2019 (Figure 4A). Many peptide therapeutics are not included in the THPdb, and the listed molecules are not limited to AMPs. Fifty-seven of the molecules are small peptide therapeutics, and most of these (37 drugs) are receptor-binding peptides that either activate or block the specific receptors to which they bind, causing a biological response. The rest of them are inhibitors of biological pathways (15 drugs), membrane-active peptides (MAPs; 4 drugs), or have other functions (1 drug) (Figure 4B). Below, we will discuss the various ways in which peptides can interact with cells to perform their therapeutic functions.

### 4.1. Receptor-Binding Peptides

Receptor-binding peptides, which include both agonists and antagonists, constitute the major category of therapeutic peptides that have been approved by the FDA. These peptides have been used as therapeutics and diagnostics for applications other than infectious diseases (see Table 1). Seventeen of the 37 FDA-approved receptor-binding peptides are insulin and its analogs, which are used for treating diabetes. Four of the 37 have been utilized as anti-cancer drugs: two for prostate cancer and two for neuroendocrine tumors. Some of the FDA-approved peptides have immune-modulating effects.

However, no receptor-binding peptide has yet received FDA-approval as an antimicrobial therapeutic [51]. Small-molecule drugs have been widely studied to modulate the immune system, e.g., drugs that interact with the toll-like receptor [52,53,54]. Given this capacity, it is possible that receptor-binding peptides could be used to treat infections by stimulating the immune system. Future directions of research may investigate the use of peptides to modulate the immune system instead of, or in addition to, killing bacteria directly.

In fact, several multifunctional AMPs have been used experimentally to modulate the immune response and kill pathogens [55,56,57]. For example, human cathelicidin LL-37 and human β defensins activate the toll-like receptor signal in the innate immune system [58,59]. Nevertheless, these peptides may be a double-edged sword in that a higher dose of AMPs (e.g., cathelicidin LL-37) or their proteolytic peptide fragments could result in off-target effects and trigger additional chronic inflammatory diseases, e.g., atopic dermatitis, rosacea, psoriasis, and hidradenitis suppurativa [60]. Clinical studies of LL-37 as a topical treatment for chronic leg ulcers has demonstrated safety [61]. LL-37 has entered phase II clinical studies for further investigation of its antimicrobial activity and its ability to modulate inflammation and the healing rate of diabetic foot ulcers (see more information on ClinicalTrials.gov website: https://clinicaltrials.gov; ClinicalTrials.gov Identifier: NCT04098562). More clinical studies of LL-37 are needed to explore the efficacy and potential side effects of this molecule [61]. In future work, synthetic peptides that have precise immunomodulatory effects together with direct antimicrobial activity may be designed as a promising route.

### 4.2. Membrane-Active Peptides (MAPs)

Five out of the seven FDA-approved AMPs are MAPs (Table 1): gramicidin, daptomycin, oritavancin, telavancin, and colistin. More specifically, gramicidin is a pore-forming peptide that forms ion channels as a transmembrane dimer. Daptomycin is a membrane lytic peptide that does not form pores but co-clusters with anionic lipids and lyses the membrane. These peptides aggregate and assemble in bacterial membranes, promoting membrane depolarization via different pathways [16,23,24,62,63,64]. Oritavancin and telavancin are dual-mechanism AMPs: they (i) inhibit bacterial cell wall synthesis and (ii) disrupt bacterial membranes. Although a few studies have suggested that these peptides have features similar to those of another pore-forming AMP, nisin [28,29,30,65], their actual mechanisms of membrane disruption remain unclear. Oritavancin and telavancin may either form membrane pores or channels, or lyse the membrane. Colistin forms pore-like aggregates in the bacterial cell membrane and disrupts the membrane; thus, it results in lytic cell death [66,67].

Membrane pore-forming AMPs constitute a large subgroup of MAPs. These peptides bind to cell membranes and spontaneously assemble in the lipid bilayer as a channel or pore-like structure, though not all are cytolytic (Figure 5). Well-known natural examples are gramicidin [68], colistin [67], melittin [69,70], maculatin [71], and alamethicin [72]. The two common models for the channel structures are barrel-stave and toroidal, depending on how the peptide interacts with the lipid headgroups [73]. These oligomeric structures can be a homogeneous population of oligomers or have diverse multimeric sizes that can conduct water and ions across the membrane. The pore size, which varies among different peptides, can be measured by several biophysical techniques and molecular dynamics simulations [44,71,74,75,76,77].

Membrane-lytic peptides, e.g., daptomycin [23,24,26,78,79], colistin [67], LL-37 [80], aurein 1.2 [81,82], and piscidin 1 [83,84], disrupt cell membranes, like detergents. These peptides accumulate on the membrane surface, carpet the membrane at a critical threshold concentration, and destabilize and permeabilize the membrane structure. Some membrane-lytic peptides, e.g., melittin and colistin, form pores at low peptide concentration and lyse the cell membrane above a threshold concentration or interact with specific membrane types [85].

Unlike receptor-binding peptides or peptide inhibitors that have specific binding targets, membrane-active peptides, whose activity is limited to specific cell membranes, are not well-defined. Their specificity is usually caused by their hydrophobic moment and electrostatic interactions, but exact mechanisms have not yet been determined [86,87,88]. These properties limit the ability to precisely tune the selectivity of membrane-active peptides toward a specific bacterial species [44,70,71,74,89,90]. Bacterial membranes are generally composed of more negative charges [91], whereas mammalian cell membranes contain abundant cholesterol and sphingomyelin, which make the membranes more rigid [92,93]. The advantage of utilizing membrane-active peptides for antibiotics is that bacteria may have less of a chance to develop drug resistance [94]. A deeper understanding of the molecular mechanisms underlying bacterial membrane disruption may enable further fine-tuning of the hydrophobic moment and charge distribution, and improvement of specificity.

### 4.3. Cell Wall-Inhibiting Peptides

Other FDA-approved AMPs are inhibitors of bacterial cell wall synthesis: vancomycin, oritavancin, dalbavancin, and telavancin. These glycopeptides bind to the D-alanyl-D-alanyl amino acids on peptidoglycan chains and prevent the incorporation of N-acetylmuramic acid and N-acetylglucosamine. Their binding blocks peptidoglycan elongation and cell wall formation, killing the bacteria [27,28,29,30,65,95].

### 4.4. Peptides Having Other Inhibitory Mechanisms

Other potential targets for inhibiting bacterial growth include DNA, RNA, and ribosomes (protein synthesis). Peptides with these mechanisms include edeine [96,97], tuberactinomycins [98], and dityromycin [99]. Edeine is an antimicrobial pentapeptide that binds to the binding site (P-site) of both 30S subunits and 70S ribosomes, thus inhibiting the binding of aminoacyl-tRNA and blocking translation initiation [96,97]. Tuberactinomycins, a group of cyclic peptides, inhibit prokaryotic protein synthesis and group I self-splicing via binding to the G-binding site and backbone of the intron RNA [98]. The antimicrobial cyclic decapeptide dityromycin has been shown to block elongation factor G (EF-G)-catalyzed translocation by disrupting the contact between EF-G and ribosomal protein S12, so that it deactivates the ribosome-EF-G complex and prevents translocation [99].

## 5. Conclusions and Future Perspectives

Peptide therapeutics have only made up a minority of all new molecular entities approved by the FDA (Figure 4A). Peptides have mostly been utilized to treat bacterial skin infections, pink eye, or wounds [100,101], e.g., Neosporin^®^ (gramicidin; manufactured by Monarch Pharmaceuticals, Inc., Bristol, TN, USA), Cubicin^®^ (daptomycin; manufactured by Merck & Co., Inc., Kenilworth, NJ, USA), Vancocin^®^HCl (vancomycin; manufactured by ANI Pharmaceuticals, Inc., Baudette, MN, USA), Orbactiv^®^ (oritavancin; manufactured by Melinta Therapeutics, Inc., New Haven, CT, USA), Dalvance^TM^ (dalbavancin; manufactured by Allergan Sales, LLC, Irvine, CA, USA), and Vibativ^®^ (telavancin; manufactured by Theravance Biopharma, Inc., San Francisco, CA, USA). Several AMPs have been approved for direct injection into the human body, e.g., Cubicin, Vancocin, Orbactiv, Dalvance, Vibativ, and Coly-Mycins, because of their longer elimination half-life (ranging from hours to days) and better pharmacokinetics [31,32,33,34] compared with gramicidin or other AMPs. However, most of these lipopeptide antibiotics (except colistin) are used for treating Gram-positive bacterial infections, and only a few of them have been administered as oral solutions or tablets because of their poor penetration of the intestinal mucosa [102]. Oral vancomycin (Vancocin) is limited to the treatment of Gram-positive bacterial infections, such as *Clostridium difficile* diarrhea and staphylococcal enteritis, because of its poor absorption and ingestion in the body and the severity of these infections. AMPs to treat infections caused by Gram-negative bacteria are clearly needed.

Although vancomycin has been approved by the FDA, several clinical studies have shown that it may cause kidney damage in some patients or at high doses. Oritavancin and dalbavancin were, in fact, developed to improve the antibacterial activity of vancomycin, so that the dose could be reduced and toxicity lowered or prevented. Although the side effects of these compounds are mild, their effectiveness against drug-resistant Gram-positive organisms and for long-term treatment remains ambiguous [103,104,105]. Telavancin, another derivative of vancomycin, is more effective for treating a range of drug-resistant Gram-positive bacteria, but it has been reported that it may induce acute kidney injury and have a higher death rate than vancomycin [106,107]. Colistin may cause damage to the kidneys and the central nervous system in adult patients, and heavy use of colistin can result in the occurrence of colistin-resistant bacteria, making it problematic for regular use [108,109,110]. Other extensively studied pore-forming AMPs, such as alamethicin and melittin, are hemolytic and cytotoxic [69,111,112]; therefore, no clinical study has been conducted (see more information on ClinicalTrials.gov website: https://clinicaltrials.gov). This suggests that controlling the selectivity, reducing the toxicity, and lowering unexpected side effects are essential to the design of AMPs as human medicines [38,49].

The seven FDA-approved AMPs are small, with a molecular weight between 1145 and 1882. They are composed of several noncanonical amino acids and have chemical modifications or cyclic structures. These features optimize their pharmacokinetics and extend their elimination half-life so that they resist enzymatic degradation. The APD contains 3156 AMPs, 98% of which were discovered in nature [13]: many, in fact, were extracted from the skin secretions of frogs [44,113,114,115] or are toxins from other species, e.g., bees, snakes, and wasps [86,116,117]. In contrast, all the FDA-approved AMPs were discovered in or derived from Gram-positive bacteria commonly found in the soil: *Brevibacillus brevis* (gramicidin), *Streptomyces roseosporus* (daptomycin), *Amycolatopsis orientalis* (vancomycin, which is the prototype of oritavancin, dalbavancin, and telavancin), and *Paenibacillus polymyxa* (colistin) [118]. This coincidence is not surprising, as soil bacteria are also the source of many conventional antibiotics.

Numerous approaches to peptide design have been introduced to make AMPs less toxic for humans while maintaining or improving their potency to eliminate bacteria [11,42], e.g., rational design [119,120], combinatorial peptide libraries [75,121], high-throughput screening [122,123], database-guided approaches [124,125], structure-function-guided design [86,126,127], and molecular dynamics simulations [44]. Three major methods to improve AMP function have been described: (i) High-throughput screening can be used to identify potential AMPs [128,129,130]. The SPOT-synthesis technique, for instance, has been applied to medium- or high-throughput screens; with this technique, peptide arrays are synthesized on a cellulose membrane, and the peptides can be easily cleaved from the support for screening. In addition, combining computer algorithms, automated synthesis, and automated screening for drug design can rapidly accelerate and reduce the cost of labor for drug discovery [131]. (ii) Conjugation of peptides to other active molecules (for example, antibodies) or incorporation of peptides into nanoparticles or dendrimers allows the advantages of both types of biomolecules to be combined and overcomes their weaknesses [132]. Synthesizing AMP polymers using dendrimer or other AMP nanoparticles to increase the local concentration of the AMP can lower the required dose and combat multidrug-resistant bacteria [133,134,135,136]. (iii) The development of in vitro tests and computational predictions, for example, testing for similarities with allergens, can help to evaluate the immunogenicity and allergenic potential of newly developed AMPs [137], e.g., the basophil activation test [138], cytokine assays [139], lymphocyte activation analysis [140], and Structural Database of Allergenic Proteins (SDAP) [141,142,143]. Although these tools cannot fully predict the allergenic effects of a new AMP in the clinic, they provide a promising preclinical tools for evaluating peptide-based drugs.

We compared the AMPs from the APD [13] with the seven FDA-approved AMPs and found that the peptide sequences and physicochemical properties (e.g., hydrophobic content and net charge) vary widely among AMPs (Figure 3). These features merit additional study to determine what contribution they make to antimicrobial activity and ultimate clinical utility. It is unclear why the development of many natural peptides has stopped during preclinical stages or why the peptides failed to show sufficient antimicrobial activity and drug-like properties in clinical studies [5,10,134,144]. Their poor performance may derive from differences between the clinical setting and their native conditions. We speculate that in nature, peptides may participate in cooperative pathways with other chemical compounds or enzymes, and these conditions may increase their potency against bacteria. Synthetic AMPs used in isolation, on the other hand, may not have equally strong antimicrobial activity.

Based on these observations, it may be important not only to design synthetic AMPs that have antimicrobial activity, but to optimize them for desirable clinical properties, such as: (i) stabilizing peptide structures and introducing non-canonical amino acids into the peptide sequences to extend their elimination half-lives; (ii) mining antimicrobial agents from Gram-positive bacteria in soil and understanding which biochemical properties make these suitable for human use; (iii) discovering AMPs that can modulate the immune system; (iv) evaluating the potential synergy of AMPs with other chemical compounds or enzymes to enhance antimicrobial activity; (v) optimizing the design of computational tools for peptide therapeutics and high-throughput screening; and (vi) developing appropriate in vitro models that mimic in vivo conditions to evaluate the allergenic effects. Overall, AMPs can be a tool against drug-resistance bacteria and a source of promising therapeutics to treat infectious diseases. More investigation in the clinical setting is suggested.

## Figures and Tables

**Figure 1 antibiotics-09-00024-f001:**
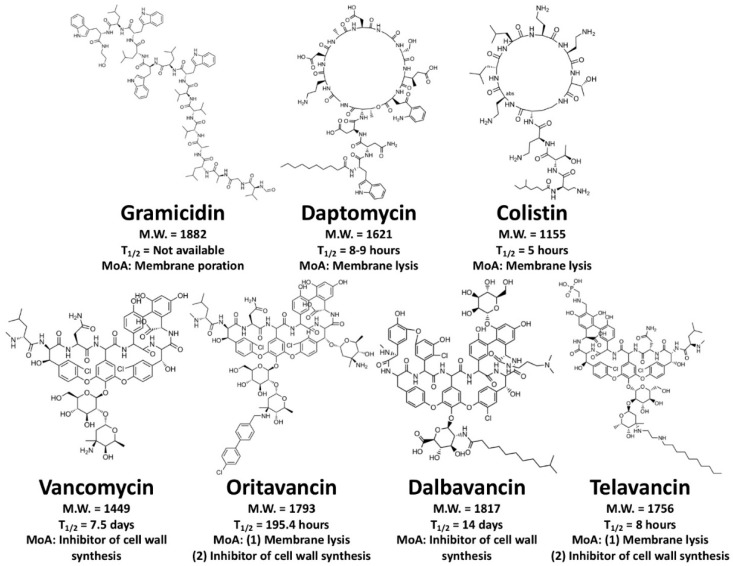
Chemical structures of seven FDA-approved AMPs. From upper left to bottom right: gramicidin (linear peptide; pore-forming peptide), daptomycin (cyclic lipopeptide; membrane-lytic peptide), colistin (cyclic lipopeptide; membrane-lytic peptide), vancomycin (lipoglycopeptide; inhibitor of cell wall synthesis), oritavancin (lipoglycopeptide; dual-mechanism: membrane-lytic peptide and inhibitor of cell wall synthesis), dalbavancin (lipoglycopeptide; inhibitor of cell wall synthesis), and telavancin (lipoglycopeptide; dual-mechanism: membrane-lytic peptide and inhibitor of cell wall synthesis). MoA indicates “mechanism of action”. T_1/2_ indicates the elimination half-life.

**Figure 2 antibiotics-09-00024-f002:**
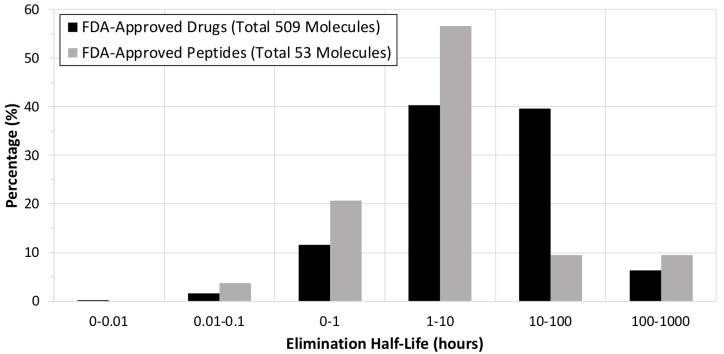
The elimination half-life (hours) of all FDA-approved drugs (509 validated molecules out of 555 molecules; black) and FDA-approved small peptides (54 validated molecules out of 57 molecules; gray). The FDA-approved drugs also include the FDA-approved small peptides. The FDA-approved small peptides are summarized in Table 1. The raw data (submission classification: Type 1—New Molecular Entity between January 1999 and December 2019) were collected from Drugs@FDA (http://www.fda.gov/drugsatfda).

**Figure 3 antibiotics-09-00024-f003:**
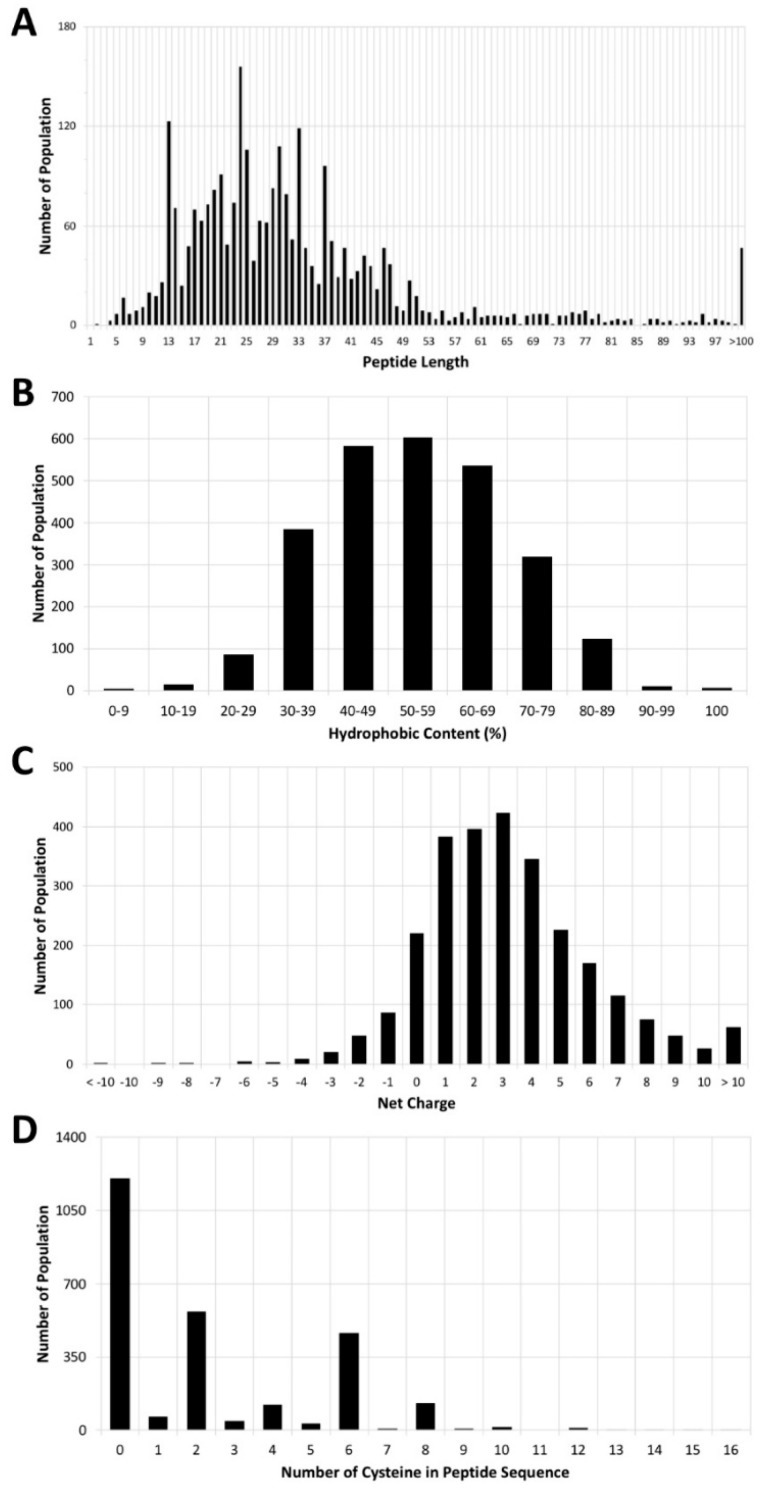
Analysis of Antimicrobial Peptide Database (APD). This analysis took into account 2700 of the AMPs listed in the APD. Distributions of (**A**) peptide length, (**B**) hydrophobic content, (**C**) net charge, and (**D**) number of cysteine residues in the peptide sequence. The raw data were collected from the APD [13].

**Figure 4 antibiotics-09-00024-f004:**
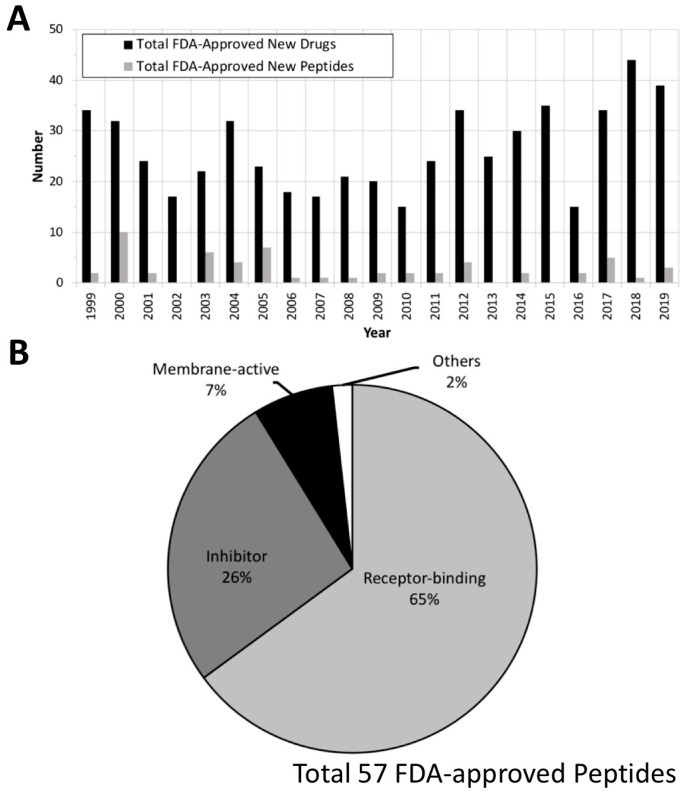
Analysis of the FDA-approved new drugs between January 1999 and December 2019. (**A**) Annual number of the total FDA-approved new drugs (black) and peptide/protein therapeutics (gray). (**B**) Known mechanisms of the FDA-approved peptide/protein therapeutics. “Inhibitor” includes membrane inhibition as well as other mechanisms of action. The raw data were collected from Drugs@FDA (http://www.fda.gov/drugsatfda).

**Figure 5 antibiotics-09-00024-f005:**
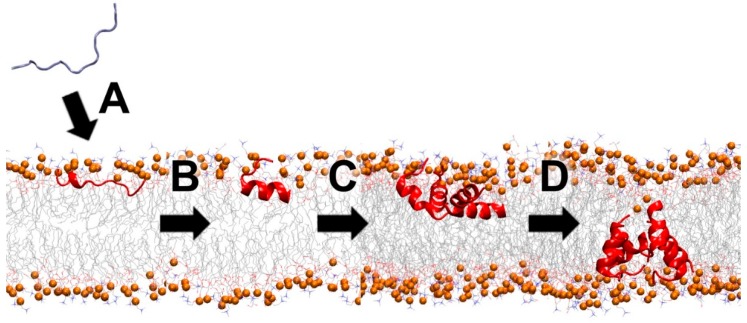
Schematic partitioning for membrane-active peptides (MAPs) that interact with the lipid bilayer. Water-soluble peptides spontaneously (**A**) bind onto the interface of the lipid bilayer, (**B**) fold and (**C**) aggregate at the membrane interface, and (**D**) form a channel and/or pore in the lipid membrane. Orange spheres represent the phosphorus atoms of the lipid headgroups.

**Table 1 antibiotics-09-00024-t001:** Summary of the small peptide (less than 50 amino acids) therapeutics approved by the FDA between January 1999 and December 2019. Raw data (submission classification: Type 1—New Molecular Entity) were collected from Drugs@FDA (http://www.fda.gov/drugsatfda) and the data were further confirmed by DrugBank [50]. “MAP” is defined as “membrane-active peptide”.

DRUG NAME	APPROVAL DATE	NDA NUMBER	ACTIVE INGREDIENTS	COMPANY	MW	PEPTIDE	APPLICATIONS	CATEGORY	ELIMINATION HALF-LIFE
**SCENESSE**	10/8/2019	210797	AFAMELANOTIDE	CLINUVEL INC	1646.85	Synthetic peptide	Prevents skin damage from the sun in patients with erythropoietic protoporphyria	Receptor binding	30 min
**GALLIUM DOTATOC GA68**	8/21/2019	210828	GALLIUM DOTATOC GA-68	UNIV IOWA HOSPS AND CLINICS PET IMAGING CENTER	1489.65	Cyclic octapeptide	Neuroendocrine tumors (NETs) diagnosis	Receptor binding	2–4 h
**VYLEESI (AUTOINJECTOR)**	6/21/2019	210557	BREMELANOTIDE ACETATE	AMAG PHARMS INC	1025.2	Cyclic heptapeptide	Hypoactive sexual desire disorder (HSDD) treatment	Receptor binding	1.9–4 h
**LUTATHERA**	1/26/2018	208700	LUTETIUM DOTATATE LU-177	AAA USA INC	1609.6	Cyclic peptide-radionuclide conjugate	Gastroenteropan-creatic neuroendocrine tumors (GEP-NETs) treatment	Receptor binding	3.5–71 h
**GIAPREZA**	12/21/2017	209360	ANGIOTENSIN II ACETATE	LA JOLLA PHARMA	1046.2	Synthetic peptide	Treatment of sepsis, septic shock, diabetes mellitus, and acute renal failure	Receptor binding	<1 min
**OZEMPIC**	12/5/2017	209637	SEMAGLUTIDE	NOVO NORDISK INC	4113.58	Chemically modified peptide	Improving glycemic control in patients with type 2 diabetes mellitus	Receptor binding	7 days
**TYMLOS**	4/28/2017	208743	ABALOPARATIDE	RADIUS HEALTH INC	3961	Synthetic peptide	Osteoporosis treatment	Receptor binding	1.7 h
**PARSABIV**	2/7/2017	208325	ETELCALCETIDE	KAI PHARMS INC	1047.5	Synthetic peptide	Treatment of secondary hyperparathyroidism	Receptor binding	3–4 days
**TRULANCE**	1/19/2017	208745	PLECANATIDE	SALIX	1682	Cyclic peptide	Chronic idiopathic constipation (CIC) treatment	Receptor binding	N/A
**ADLYXIN**	7/27/2016	208471	LIXISENATIDE	SANOFI-AVENTIS US	4858.5	Synthetic peptide	Type 2 diabetes mellitus (T2DM) treatment	Receptor binding	1–3.5 h
**NETSPOT**	6/1/2016	208547	GALLIUM DOTATATE GA-68	AAA USA INC	1435.6	Cyclic peptide-radionuclide conjugate	Neuroendocrine tumors (NETs) diagnosis	Receptor binding	1 h
**ORBACTIV**	8/6/2014	206334	ORITAVANCIN DIPHOSPHATE	MELINTA THERAP	1989.09	Lipoglycopeptide	Treatment of complicated skin and skin structure infections (cSSSI) caused by gram-positive bacteria	MAP	195.4 h
**DALVANCE**	5/23/2014	021883	DALBAVANCIN HYDROCHLORIDE	ALLERGAN SALES LLC	~1800	Lipoglycopeptide	Treatment of complicated skin and skin structure infections (cSSSI) caused by gram-positive bacteria	Inhibitor	346 h
**GATTEX KIT**	12/21/2012	203441	TEDUGLUTIDE RECOMBINANT	NPS PHARMS INC	3752	Glucagon-like peptide-2	Short bowel syndrome (SBS) treatment	Receptor binding	1.3–2 h
**SIGNIFOR**	12/14/2012	200677	PASIREOTIDE DIASPARTATE	NOVARTIS	1313.41	Cyclohexapeptide	Treatment of Cushing’s disease	Receptor binding	10–13 h
**LINZESS**	8/30/2012	202811	LINACLOTIDE	ALLERGAN SALES LLC	1526.8	Cyclic peptide	Treatment of irritable bowel syndrome	Receptor binding	N/A
**KYPROLIS**	7/20/2012	202714	CARFILZOMIB	ONYX THERAP	719.9	Modified tetrapeptidyl epoxide	Multiple myeloma treatment	Inhibitor	≤1 h
**FIRAZYR**	8/25/2011	022150	ICATIBANT ACETATE	SHIRE ORPHAN THERAP	1304.5	Synthetic peptide	Treatment of angioedema, liver disease, burns, and burn infections	Receptor binding	1.4 h
**INCIVEK**	5/23/2011	201917	TELAPREVIR	VERTEX PHARMS	679.85	Chemically modified peptide	Treatment of chronic Hepatitis C	Inhibitor	4–11 h
**EGRIFTA**	11/10/2010	022505	TESAMORELIN ACETATE	THERATECHNOLOGIES	5135.9	Synthetic peptide	Human immunodeficiency virus (HIV) treatment	Receptor binding	26–38 min
**VICTOZA**	1/25/2010	022341	LIRAGLUTIDE RECOMBINANT	NOVO NORDISK INC	3751.2	Synthetic peptide	Improving glycemic control in patients with type 2 diabetes mellitus	Receptor binding	13 h
**ISTODAX**	11/5/2009	022393	ROMIDEPSIN	CELGENE	540.71	Bicyclic peptide	Treatment of cutaneous T-cell lymphoma (CTCL) and/or peripheral T-cell lymphoma (PTCL)	Inhibitor	3 h
**VIBATIV**	9/11/2009	022110	TELAVANCIN HYDROCHLORIDE	CUMBERLAND PHARMS	1755.6	Lipoglycopeptide	Treatment of complicated skin and skin structure infections (cSSSI) caused by Gram-positive bacteria	MAP	8 h
**FIRMAGON**	12/24/2008	022201	DEGARELIX ACETATE	FERRING	1632.3	Synthetic peptide	Prostate cancer treatment	Receptor binding	53 h
**SOMATULINE DEPOT**	30/08/2007	022074	LANREOTIDE ACETATE	IPSEN PHARMA	1096.34	Cyclical octapeptide	Treatment of neuroendocrine tumors (NETs) and acromegaly	Receptor binding	22 days
**ERAXIS**	2/17/2006	021632	ANIDULAFUNGIN	VICURON	1140.3	Lipopeptide	Anti-fungal drug	Inhibitor	40–50 h
**LEVEMIR**	6/16/2005	021536	INSULIN DETEMIR RECOMBINANT	NOVO NORDISK INC	5916.9	A long-acting basal insulin analog	Treatment of hyperglycemia caused by type 1 and type 2 diabetes	Receptor binding	57 h
**LEVEMIR FLEXPEN**	6/16/2005	021536	INSULIN DETEMIR RECOMBINANT	NOVO NORDISK INC	5916.9	A long-acting basal insulin analog	Treatment of hyperglycemia caused by type 1 and type 2 diabetes	Receptor binding	5–7 h
**LEVEMIR FLEXTOUCH**	6/16/2005	021536	INSULIN DETEMIR RECOMBINANT	NOVO NORDISK INC	5916.9	A long-acting basal insulin analog	Treatment of hyperglycemia caused by type 1 and type 2 diabetes	Receptor binding	5–7 h
**LEVEMIR INNOLET**	6/16/2005	021536	INSULIN DETEMIR RECOMBINANT	NOVO NORDISK INC	5916.9	A long-acting basal insulin analog	Treatment of hyperglycemia caused by type 1 and type 2 diabetes	Receptor binding	5–7 h
**LEVEMIR PENFILL**	6/16/2005	021536	INSULIN DETEMIR RECOMBINANT	NOVO NORDISK INC	5916.9	A long-acting basal insulin analog	Treatment of hyperglycemia caused by type 1 and type 2 diabetes	Receptor binding	5–7 h
**BYETTA**	4/28/2005	021773	EXENATIDE SYNTHETIC	ASTRAZENECA AB	4186.6	Synthetic peptide	Improving glycemic control in patients with type 2 diabetes mellitus	Receptor binding	2.4 h
**SYMLIN**	3/16/2005	021332	PRAMLINTIDE ACETATE	ASTRAZENECA AB	3949.4	Peptide hormone	Treatment of type 1 and type 2 diabetes mellitus	Receptor binding	48 min
**PRIALT**	12/28/2004	021060	ZICONOTIDE ACETATE	TERSERA THERAPS LLC	2639	Synthetic peptide	Chronic pain treatment	Inhibitor	2.9–6.5 h
**APIDRA**	4/16/2004	021629	INSULIN GLULISINE RECOMBINANT	SANOFI AVENTIS US	5823	Human insulin analog	Treatment of hyperglycemia caused by type 1 and type 2 diabetes	Inhibitor	13–86 min
**APIDRA SOLOSTAR**	4/16/2004	021629	INSULIN GLULISINE RECOMBINANT	SANOFI AVENTIS US	5823	Human insulin analog	Treatment of hyperglycemia caused by type 1 and type 2 diabetes	Inhibitor	13–86 min
**CHIRHOSTIM**	4/9/2004	021256	SECRETIN SYNTHETIC HUMAN	CHIRHOCLIN	3039.44	Gastrointestinal peptide hormone	(1) Pancreatic secretions to aid in the diagnosis of pancreatic exocrine dysfunction(2) Gastrin secretion to aid in the diagnosis of gastrinoma(3) Pancreatic secretions to facilitate the identification of the ampulla of Vater and accessory papilla during ERCP	Inhibitor	45 min
**PLENAXIS**	11/25/2003	021320	ABARELIX	SPECIALTY EUROPEAN	1416.06	Synthetic peptide	Prostate cancer treatment	Inhibitor	13.2 days
**CUBICIN**	9/12/2003	021572	DAPTOMYCIN	CUBIST PHARMS LLC	1620.67	Cyclic lipopeptide	Treatment of complicated skin and skin structure infections (cSSSI) caused by Gram-positive bacteria	MAP	8.1–9 h
**CUBICIN RF**	9/12/2003	021572	DAPTOMYCIN	CUBIST PHARMS LLC	1620.67	Cyclic lipopeptide	Treatment of complicated skin and skin structure infections (cSSSI) caused by Gram-positive bacteria	MAP	8.1–9 h
**REYATAZ**	6/20/2003	021567	ATAZANAVIR SULFATE	BRISTOL MYERS	704.9	Azapeptide	Human immunodeficiency virus (HIV) treatment	Inhibitor	6.5–7.9 h
**VELCADE**	5/13/2003	021602	BORTEZOMIB	MILLENNIUM PHARMS	384.24	Chemically modified peptide	Multiple myeloma treatment	Inhibitor	9–15 h
**FUZEON**	3/13/2003	021481	ENFUVIRTIDE	ROCHE	4492	Synthetic peptide	Human immunodeficiency virus (HIV) treatment	Inhibitor	3.8 h
**NATRECOR**	8/10/2001	020920	NESIRITIDE RECOMBINANT	SCIOS LLC	3464	Cyclic peptide	Acute decompensated heart failure (ADHF) treatment	Receptor binding	18 min
**CANCIDAS**	1/26/2001	021227	CASPOFUNGIN ACETATE	MERCK	1213.42	Cyclic lipopeptide	Anti-fungal drug	Inhibitor	9–11 h
**ANGIOMAX**	12/15/2000	020873	BIVALIRUDIN	SANDOZ INC	2180	Synthetic peptide	Treatment of heparin-induced thrombocytopenia and for the prevention of thrombosis	Inhibitor	22 min–3.5 h
**CETROTIDE**	8/11/2000	021197	CETRORELIX	EMD SERONO INC	1431.06	Synthetic peptide	For prevention of premature ovulation in women undergoing fertility treatments with controlled ovulation	Receptor binding	~62.8 h
**TRELSTAR**	6/15/2000	020715	TRIPTORELIN PAMOATE	ALLERGAN SALES LLC	1699.9	Synthetic peptide	Prostate cancer treatment	Receptor binding	6 min–3 h
**NOVOLOG**	6/7/2000	020986	INSULIN ASPART RECOMBINANT	NOVO NORDISK INC	5825.8	Peptide hormone	Treatment of hyperglycemia caused by type 1 and type 2 diabetes	Receptor binding	81 min
**NOVOLOG FLEXPEN**	6/7/2000	020986	INSULIN ASPART RECOMBINANT	NOVO NORDISK INC	5825.8	Peptide hormone	Treatment of hyperglycemia caused by type 1 and type 2 diabetes	Receptor binding	81 min
**NOVOLOG FLEXTOUCH**	6/7/2000	020986	INSULIN ASPART RECOMBINANT	NOVO NORDISK INC	5825.8	Peptide hormone	Treatment of hyperglycemia caused by type 1 and type 2 diabetes	Receptor binding	81 min
**NOVOLOG INNOLET**	6/7/2000	020986	INSULIN ASPART RECOMBINANT	NOVO NORDISK INC	5825.8	Peptide hormone	Treatment of hyperglycemia caused by type 1 and type 2 diabetes	Receptor binding	81 min
**NOVOLOG PENFILL**	6/7/2000	020986	INSULIN ASPART RECOMBINANT	NOVO NORDISK INC	5825.8	Peptide hormone	Treatment of hyperglycemia caused by type 1 and type 2 diabetes	Receptor binding	81 min
**LANTUS**	4/20/2000	021081	INSULIN GLARGINE RECOMBINANT	SANOFI AVENTIS US	6063	Peptide hormone	Treatment of hyperglycemia caused by type 1 and type 2 diabetes	Receptor binding	N/A
**LANTUS SOLOSTAR**	4/20/2000	021081	INSULIN GLARGINE RECOMBINANT	SANOFI AVENTIS US	6063	Peptide hormone	Treatment of hyperglycemia caused by type 1 and type 2 diabetes	Receptor binding	N/A
**NEO TECT KIT**	8/3/1999	021012	TECHNETIUM TC-99M DEPREOTIDE	CIS BIO INTL SA	486.14	Cyclic peptide	(1) Detecting coronary artery disease(2) Evaluating myocardial function	Others	6.02 h
**GANIRELIX ACETATE**	7/29/1999	021057	GANIRELIX ACETATE	ORGANON USA INC	1570.35	Peptide hormone	For inhibition of premature LH surges in women undergoing controlled ovarian hyperstimulation	Receptor binding	16.2 h

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
