# Peer review of "Development and Challenges of Antimicrobial Peptides for Therapeutic Applications"

_antibiotics, 2020, doi:10.3390/antibiotics9010024_

Round 1

Reviewer 1 Report

This paper by Chen and Lu presents an in-depth study of antimicrobial peptides discovered over the past 20 years and sets them off against those that have achieved FDA approval. As there currently is a large discovery void in antibiotic drugs while resistant bacteria are on the rise, this review provides important insight into potential new design criteria for AMPs. Although the review gives a good survey of the field, at times too much assumption of background knowledge of the reader is made. As this paper is to be part of a special issue, I believe it is important some basic concepts are explained or illustrated as those new to AMPs are likely to use this issue to gain an understanding, not just of the current state of the field, but also of the field as a whole.

To accommodate for this, I suggest the following changes alongside some minor corrections:

page 2, line 45: Change 'unknown discovery void' to 'unknown cause of this discovery void'

page 2, line 63: Correction to references. Now (16) (17-21) should become (16-21)

page 2, lines 67-70: There is a lot of information in this sentence that is not crucial. I suggest re-formatting along the lines of 'Daptomycin (also known as LY146032 (25)) and its product Cubicin (manufactured by Cubist Pharmaceutics, now Merck & Co.) were approved in 2003 by the FDA to treat or prevent infectious diseases (26).

Page 3, line 94: Change 'and the small peptides' to 'and for the small peptides'

Page 3, lines 95-96: It is not clear how this statement follows from the preceding text. I suggest creating a new paragraph (potentially before line 90 on this page) discussing the problems that arise from short half-life of peptides and a few ways in which these can be improved.

Page 4, lines 128-130: Please state more clearly how this analysis is different from the ones discussed before e.g. that it is not limited to antimicrobial peptides.

Page 4, line 137: the two sections are disconnected.  Give a set-up for the discussion that follows i.e. ' below we will discuss the various ways in which peptides can interact with cells to perform their therapeutic functions'.

Page 4, lines 145-146: For an audience that might be new to the field of AMPs it might be worth explaining why these other peptides could potentially be used for antimicrobial treatment. What precedent is there for this?

Page 4, line 154: A more general suggestion for this section; a figure illustrating the different mechanisms through which antimicrobial peptides can work will be helpful for a broader audience.

Page 5, lines 188-190: Please restructure this sentence so effect follows cause and expand the discussion of how improvement of specificity can prevent side effects with 2-3 sentences.

Page 6, line 215: Change 'This discovery void' to 'The cause of this discovery void'.

Page 6, lines 237-239: This discussion feels without conclusion, please state whether Dalbavancin achieved its aim of functioning at a lower dosage and less toxicity.

Page 7, paragraph from line 264 to 278: This requires some restructuring. Suggest to follow this layout: Two methods towards improving AMP function are presented (1) use of high-throughput screening and optimization of this methodology and (2) conjugation of peptides to (i) other active molecules (antibodies) or (ii) incorporation of peptides into nano-particles or dendrimers.

Page 8, lines 280-281: This is a rather tame conclusion that in my opinion takes away from the great insight presented in this review. Can this be tied back to the extensive survey the authors have performed? What are key findings from the FDA approved peptides that can be used as guidelines for design of new AMPs?

Author Response

Response to Reviewer 1 Comments

We carefully went through your comments and revised the content. Thank you so much for your thorough review and many valuable suggestions.

This paper by Chen and Lu presents an in-depth study of antimicrobial peptides discovered over the past 20 years and sets them off against those that have achieved FDA approval. As there currently is a large discovery void in antibiotic drugs while resistant bacteria are on the rise, this review provides important insight into potential new design criteria for AMPs. Although the review gives a good survey of the field, at times too much assumption of background knowledge of the reader is made. As this paper is to be part of a special issue, I believe it is important some basic concepts are explained or illustrated as those new to AMPs are likely to use this issue to gain an understanding, not just of the current state of the field, but also of the field as a whole.

To accommodate for this, I suggest the following changes alongside some minor corrections:

Point 1: page 2, line 45: Change 'unknown discovery void' to 'unknown cause of this discovery void'

Response 1: We have corrected it.

Point 2: page 2, line 63: Correction to references. Now (16) (17-21) should become (16-21)

Response 2: We have corrected it.

Point 3: page 2, lines 67-70: There is a lot of information in this sentence that is not crucial. I suggest re-formatting along the lines of 'Daptomycin (also known as LY146032 (25)) and its product Cubicin (manufactured by Cubist Pharmaceutics, now Merck & Co.) were approved in 2003 by the FDA to treat or prevent infectious diseases (26).

Response 3: We have corrected it.

Point 4: Page 3, line 94: Change 'and the small peptides' to 'and for the small peptides'

Response 4: We have corrected it.

Point 5: Page 3, lines 95-96: It is not clear how this statement follows from the preceding text. I suggest creating a new paragraph (potentially before line 90 on this page) discussing the problems that arise from short half-life of peptides and a few ways in which these can be improved.

Response 5: We have introduced a new paragraph to connect the texts.

Point 6: Page 4, lines 128-130: Please state more clearly how this analysis is different from the ones discussed before e.g. that it is not limited to antimicrobial peptides.

Response 6: We have specified it.

Point 7: Page 4, line 137: the two sections are disconnected.  Give a set-up for the discussion that follows i.e. ' below we will discuss the various ways in which peptides can interact with cells to perform their therapeutic functions'.

Response 7: We have corrected it.

Point 8: Page 4, lines 145-146: For an audience that might be new to the field of AMPs it might be worth explaining why these other peptides could potentially be used for antimicrobial treatment. What precedent is there for this?

Response 8: We have described it in the text.

Point 9: Page 4, line 154: A more general suggestion for this section; a figure illustrating the different mechanisms through which antimicrobial peptides can work will be helpful for a broader audience.

Response 9: We have made a figure to illustrate how membrane-active peptides work.

Point 10: Page 5, lines 188-190: Please restructure this sentence so effect follows cause and expand the discussion of how improvement of specificity can prevent side effects with 2-3 sentences.

Response 10: We have expanded the discussions.

Point 11: Page 6, line 215: Change 'This discovery void' to 'The cause of this discovery void'.

Response 11: We have corrected it.

Point 12: Page 6, lines 237-239: This discussion feels without conclusion, please state whether Dalbavancin achieved its aim of functioning at a lower dosage and less toxicity.

Response 12: We have addressed it in the text.

Point 13: Page 7, paragraph from line 264 to 278: This requires some restructuring. Suggest to follow this layout: Two methods towards improving AMP function are presented (1) use of high-throughput screening and optimization of this methodology and (2) conjugation of peptides to (i) other active molecules (antibodies) or (ii) incorporation of peptides into nano-particles or dendrimers.

Response 13: We have followed the layout and re-formatted the content.

Point 14: Page 8, lines 280-281: This is a rather tame conclusion that in my opinion takes away from the great insight presented in this review. Can this be tied back to the extensive survey the authors have performed? What are key findings from the FDA approved peptides that can be used as guidelines for design of new AMPs?

Response 14: We have revised the conclusion and highlighted what we learned from the FDA approved AMPs.

Reviewer 2 Report

The reviewed manuscript entitled “Development and Challenges of Antimicrobial Peptides for Therapeutic Applications” presents very important issues related to the development of AMPs. The manuscript is well written and clearly presented. The work can be of great interest to scientists and clinicians. However, the work should be extended to some issues. The manuscript lacks paying attention to an immunogenicity and allergic potential of newly developed peptides e.g. on the basis of mathematical calculations of allergens similarities (data bases) and data acquired in in vitro tests (Lymphocytes, basophils). Moreover, the work has not sufficiently mentioned the strategy of increasing the peptides stability. I suggest to publish this work after minor revision and consideration the above mentioned issues.

Author Response

Response to Reviewer 2 Comments

Point 1: The reviewed manuscript entitled “Development and Challenges of Antimicrobial Peptides for Therapeutic Applications” presents very important issues related to the development of AMPs. The manuscript is well written and clearly presented. The work can be of great interest to scientists and clinicians. However, the work should be extended to some issues. The manuscript lacks paying attention to an immunogenicity and allergic potential of newly developed peptides e.g. on the basis of mathematical calculations of allergens similarities (data bases) and data acquired in in vitro tests (Lymphocytes, basophils). Moreover, the work has not sufficiently mentioned the strategy of increasing the peptides stability. I suggest to publish this work after minor revision and consideration the above mentioned issues.

Response 1: We carefully went through your comments and revised the content. We have extended the discussions and made this manuscript clear and comprehensive. Thank you so much for spending your valuable time on this paper.